# Reversing Structural Pattern Learning with Biologically Inspired Knowledge Distillation for Spiking Neural Networks

## ABSTRACT

Spiking neural networks (SNNs) have superb characteristics in sensory information recognition tasks due to their biological plausibility. However, the performance of some current spiking-based models is limited by their structures which means either fully connected or too-deep structures bring too much redundancy. This redundancy from both connection and neurons is one of the key factors hindering the practical application of SNNs. Although Some pruning methods were proposed to tackle this problem, they normally ignored the fact the neural topology in the human brain could be adjusted dynamically. Inspired by this, this paper proposed an evolutionary-based structure construction method for constructing more reasonable SNNs. By integrating the knowledge distillation and connection pruning method, the synaptic connections in SNNs can be optimized dynamically to reach an optimal state. As a result, the structure of SNNs could not only absorb knowledge from the teacher model but also search for deep but sparse network topology. Experimental results on CIFAR100, Tiny-imagenet and DVS-Gesture show that the proposed structure learning method can get pretty well performance while reducing the connection redundancy. The proposed method explores a novel dynamical way for structure learning from scratch in SNNs which could build a bridge to close the gap between deep learning and bio-inspired neural dynamics.

## KEYWORDS

Spiking Neural Networks, Knowledge Distillation, Brain-Inspired Models

## 1 INTRODUCTION

Spiking Neural Networks (SNNs) are supposed to be one of the efficient computational models because of their biological plausibility, especially since the structures are dynamically malleable. Cognitive activities can be realized with the help of the complex structures of the human brain which is composed of billions of neurons and more neural connections between them. Similar to biological neural networks, SNNs transfer and process information via binary spikes. During the flow of the information, those fired spikes are transmitted on the synaptic connection between neurons with the structural plasticity rules.

Compared to conventional artificial neural network models including convolutional neural networks (CNNs), definitely, SNNs are more biologically inspired and energy efficient. However, there is one fact that cannot be ignored the SNNs cannot achieve nearly as good performance as ANNs did. One of the key factors that we think some of the structures of current spiking based models are limited by training methods, that is SNNs cannot leverage the global backpropagation (BP) rules directly as CNNs did. This defect directly leads to unreasonable structures in SNNs. Besides, whether structures of CNNs or SNNs are both fixed, unlike biological neural systems, the structures and topologies should have been dynamical which means the connections could be discarded as needed.

Aiming at tackling this problem, some studies focus on constructing more efficient structures to improve the efficiency of SNNs. Researchers proposed an approximate BP method named STBP (Spatio-temporal backpropagation) for training high-performance SNNs [25]. To avoid training SNNs directly, some studies [6, 11] proposed ANN-to-SNN get parameters from trained ANNs, then map them to SNNs. Although these methods could construct pretty deep structures, they bring additional computational power. Moreover, these BP based learning rules can only match the fixed structure, if changing the network topology, it would be retrained from scratch.

Aiming at constructing more biologically flexible SNNs, this paper proposed biologically inspired structure learning methods with reverse knowledge distillation (KD). Based on the proposed training method, the wanted student SNN models could learn rich information from teacher ANN models [26]. Compared to traditional KD methods, one of the key differences of the proposed re-KD method for SNN training is that we think the structures play an important role in the training process, they are not only the final results, they could instruct themselves to train. Not limited to label smoothing regularization [29], this paper proved that the proposed re-KD method could build more robust structures of SNNs. We evaluated the proposed methods on several pattern recognition datasets (CIFAR100, Tiny-imagenet and DVS-Gesture) experimental results show that the proposed methods can not only get good recognition performance but also show robust generation ability in time-series dynamic environments. The main contributions in this paper are summarized as follows:

- This paper proposed reverse KD methods for constructing efficient structures of spiking neural networks. The proposed methods are emphasized to circumvent the no-differentiable of the spikes when using BP rules directly.
- The proposed methods let sparse structures models as teachers help construct robust student SNN models. Besides, this paper provides a brandnew teacher-free KD method which could help student SNN models absorb useful information when the teacher is default.

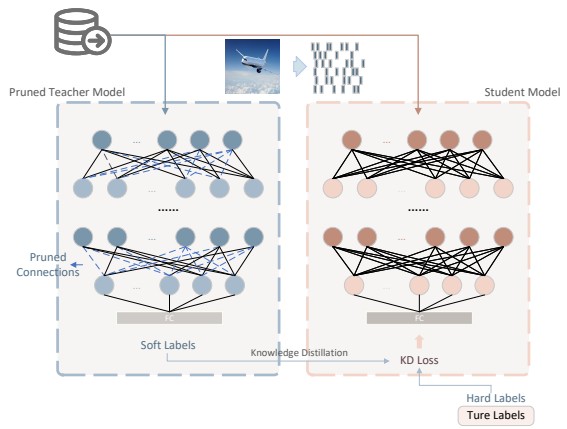
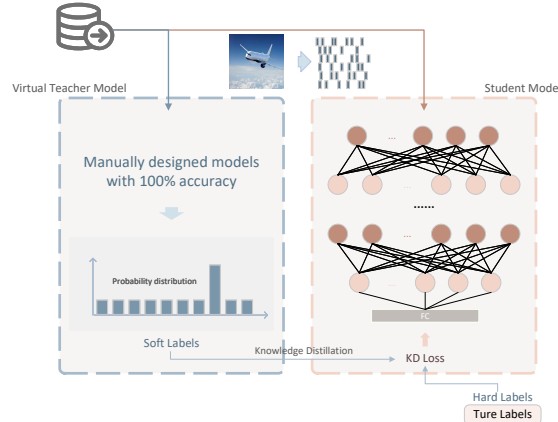

**Figure 1: The schematic illustration of the reverse-KD framework between teacher SNN and student SNN. (a) When the structure of teacher SNN is sparse, this kind of re-KD is named sparse-KD. (b) When teacher SNN is default as a probability distribution manually, this kind of re-KD is named teacher default-KD.**

- Experimental evaluations showed that the proposed re-KD method could facilitate the performance of SNNs. Furthermore, this kind of KD construction for deep SNNs could allow teacher and student models to be homogeneous or heterogeneous, which shows great potential on neuromorphic hardware platforms.

## 2 RELATED WORK AND MOTIVATION

There lacking suitable structures for deep SNNs, which caused difficulties in training deep SNNs using BP rules directly. Based on the KD method, this paper proposed reverse-KD method to further facilitate the brain-inspired characteristics of dynamic spiking signals. The proposed reverse-KD methods include two ways, one is when the structures of the teacher model are sparse named sparse-KD, and the other is when the teacher is the default during KD training process named teacher default-KD. For further play and to verify the effectiveness of the proposed method, this paper embedded surrogated gradient method into the proposed method to train the deep SNNs.

### 2.1 The structures of deep SNNs

Compared to the structures in deep learning field, those of SNNs are various because there are no unified learning rules for deep SNNs training as BP did in deep learning. Deng et al. [5], Xu et al. [27] proposed temporal efficient training methods to build deep SNNs. Chen et al. [2] dug out the state transition of dendritic spines to improve the training efficiency of sparse SNNs. Some studies focused on the activation function in SNNs [1, 31] to build high brain-inspired spiking neurons and neural circuits.

Considering more detailed differences between ANNs and SNNs [3, 19], some researchers are committed to improving the network structure so that to make SNNs could be applied to detection [13] and dynamic time-series data processing [12]. Although they can build efficient deep structures of SNNs, they also bring huge power consumption and additional computing resources. Especially, when they used ANN-to-SNN conversion [1, 21]. Besides, these structure

construction methods have overlooked the flexibility of structure creation which means if the task changed, you must train another brand-new SNN from scratch.

### 2.2 Surrogate gradient method for SNN training

Different from ANN-to-SNN conversion [7, 11], because of the huge success that BP gets in deep CNNs, some studies also want to utilize the BP-based rules to train deep SNNs. Neftci et al. [18] firstly introduced surrogated gradient training method to mimic the backpropagation process in CNNs. These types of methods aimed at the no-differentiable problem in binary spikes, by applying surrogate gradient to make spikes become differentiable so that the corresponding weights can be trained globally.

Zenke and Vogels [30] analysed the robustness of surrogate gradient training in SNNs and instilled the complex function in SNNs. Combing the characteristic of membrane potential, some scholars want to incorporate learnable membrane time constant to enhance the surrogated gradient training [6]. Others made a further improvement to make it friendly to neuromorphic on-chip hardware systems [23].

Although these surrogated training methods consume little power consumption compared to ANN-to-SNN conversion, they still bring additional computational resources. More importantly, they still make the structures of SNNs fixed and cannot change with the change of tasks. To build more flexible structures of SNNs, combing with surrogated-gradient training, some studies proposed knowledge distillation-based methods to build more efficient SNNs [15, 16, 24]. They normally set a powerful ANN as a teacher and a shallow SNN as a student, it will make unrealistic assumptions and introduce more computing consumption when training a strong ANN additionally.

### 2.3 Motivation

Aiming at tackling the aforementioned problems in constructing efficient deep SNN models, this paper proposed a reverse KD method to construct deep SNNs. Through re-think the original KD in SNNs,

this paper makes a systematic analysis of the knowledge transfer between teacher and student.

This paper not only focused on the efficient structures of SNNs, but also care about the power consumption brought by KD process. Specifically, we build two re-KD methods for deep SNN construction, one is sparse-KD and the other is teacher default-KD. One of the key innovations is that we think the teacher is not always strong meanwhile, the student is not always weak. By building the relationship between weak teachers and strong students, this paper rethinks the KD process in SNN training and proved that based on the proposed method, the performance would be improved meanwhile the power consumption decreases.

## 3 METHOD

In order to construct a bio-inspired training method for combining structural learning and knowledge distillation for SNNs, this paper proposes the re-KD method. Our method presents two approaches for knowledge distillation: sparse-KD and teacher default-KD. This knowledge distillation framework explores a method of knowledge distillation that utilizes a network with a sparse structure or a virtual network as the teacher network, which is different from conventional knowledge distillation methods.

### 3.1 The framework of reverse knowledge distillation

**Spiking neuron model.** We use IF (Integrate-And-Fire Models) spiking neuron models as the basic unit of the network. The IF neuron model is one of the commonly used spiking neuron models. It stimulates the process of action potential in biological neurons, where it receives spiking stimulation from presynaptic neurons and the state of membrane potential changes. The neural dynamics equation of the IF neuron can be represented by Eq. (1).

$$\frac{\mathrm{d}V(t)}{\mathrm{d}t} = V(t) + X(t) \qquad (1)$$

where $V(t)$ is the membrane potential of IF neuron. $X(t)$ denotes the current input at time $t$. Therefore, the current membrane potential of the IF neuron can be expressed as Eq. (2).

$$V(t) = f(V(t-1), X(t)) = V(t-1) + X(t) \qquad (2)$$

When the membrane potential reaches the threshold, the neuron will fire a spike, and then its membrane potential will be reset to the reset voltage $V_{\mathrm{reset}}$. In addition to this, the framework trains SNN based on gradient surrogates.

**Overall framework of the reverse knowledge distillation method.** Re-KD explores a novel way of knowledge distillation, which combines bio-inspired structure and distillation. A reverse knowledge distillation approach is adopted to train the SNN student network. As shown in Section 1, the pruned SNN model is used as a teacher network to guide the training of the student network, which can reduce the interference in the hidden information of the teacher network to better train the student network. As shown in Section 1, a 100% accuracy virtual teacher network is designed to avoid the impact of wrong classification better. Then this framework adopts a response-based knowledge distillation method to train a student network.

### 3.2 Sparse knowledge distillation

**Teacher sparsified.** This method uses a sparse network as the teacher network which is called sparse-KD. We use a weight-based pruning method to obtain a sparse network structure as the teacher model to guide the training of the student model. This sparse method prunes a certain proportion of connections with low-weight values from a pre-trained SNN model. There is often redundancy in the connections within a network structure, which is similar to that of a biological neural network. Removing these redundant connections makes the network structure more robust. First, we train an SNN model and fix the weights of the model. Then, we sort the weights of all connections in the convolutional layers of the model and prune the weights with the smallest values according to a certain proportion. This method sets a mask that is multiplied by the weights, with the elements in the corresponding positions of the mask matrix set to 0 for the connections with weights that need to be pruned. The weight and pruning mask in $l$ layer is $W^l$ and $M^l$ and the weight after pruning can be expressed as Eq. (3):

$$W^l_{\mathrm{pruned}} = W^l \odot M^l \qquad (3)$$

**Loss function.** This method uses a knowledge distillation method based on response, where the final layer output of the teacher network is used to guide the training of the student network. In this case, the teacher network refers to a pre-trained SNN model that has been pruned. The learning objectives for the student network are divided into soft labels and true labels. Soft labels refer to the output of the final layer of the teacher network, which contains hidden knowledge. In order to increase the entropy of the probability distribution of the network's output, a temperature parameter $T$ is introduced to smooth this probability distribution. This allows for better learning of the hidden similarity information in the probability distribution of the teacher network's output. The probability distribution is represented by $Z_i$. The flatten output probability distribution $q_i$ can be expressed as Eq. (4):

$$q_i = \frac{\exp(Z_i/T)}{\sum \exp(Z_j/T)} \qquad (4)$$

The loss function for knowledge distillation during training of a student network consists of two parts. One is the loss calculated using cross-entropy between the output of the student network and the true labels. $Q_s$ and $Q_t$ is the probability distribution of the student and teacher network's output. The other is the loss calculated using KL divergence between the output of the student network and the soft labels. In the $L_{soft}$ function, the probability distributions of the outputs of the student network and the teacher network are both flattened by parameter temperature. The soft loss is computed by the simplified KL divergence. In the $L_{hard}$ function, the loss is computed by the CrossEntropy. The student network can be trained using the loss function in Eq. (5), as referenced in paper [17].

$$L_{sparse-KD} = \alpha * L_{soft}(Q_s, Q_t) + (1 - \alpha) * L_{hard}(Q_s, y_{\mathrm{true}}) \qquad (5)$$

where $\alpha$ controls the importance of the two parts of the loss function. $y_{\mathrm{true}}$ denotes the true labels.

**Training algorithm.** The first step is to train an SNN model and then prune the model based on the weights and save it as the teacher network. In the second step, we use an SNN model with the same structure as the teacher network as the student network. Then we use knowledge distillation based on response to train the student network. Through this method, the clearer hidden knowledge in the pruned teacher network is utilized to guide the training of the student network.

### 3.3 Teacher default knowledge distillation

**Teacher default.** This method uses a probability distribution designed manually as the teacher network, which allows the teacher network to have an accuracy of 100%. In normal knowledge distillation, the accuracy of the teacher network is usually not 100%, so there may be some interference in its hidden knowledge. Using a pruned teacher network for knowledge distillation can yield a more robust student network. In order to better guide the learning of the student network, a 100% accurate virtual teacher network can be designed. Assuming there are $C$ classification categories, this probability distribution sets the probability of the correct label $\alpha$ ($\alpha > 0.9$), and the probability distribution of this virtual teacher network can be represented by Eq. (6).

$$p(c) = \begin{cases} \alpha & \text{if } c = t \\ \dfrac{1 - \alpha}{C - 1} & \text{if } c \neq t \end{cases} \quad (6)$$

where $c$ represents each category and $t$ is the correct category. The probability distribution $p(c)$ can be viewed as the probability distribution of the output of this virtual teacher network.

**Loss function.** This method is similar to the response-based knowledge distillation method. The loss function of the student network training is divided into two parts. One is the cross-entropy loss between the output of the student network and the true label. The other is the KL divergence loss between the output and the virtual label. The probability distribution of this virtual teacher network can be flattened using the parameter temperature to obtain the soft label. The two parts are summed to obtain the loss function of this knowledge distillation, as shown in Eq. (7).

$$L_{default-KD} = \alpha * L_{virtual}\left(Q_s, Q_v\right) + (1 - \alpha) * L_{hard}\left(Q_s, y_{\text{true}}\right) \quad (7)$$

where $Q_s$ is the output of student network. $Q_v$ denotes the probability distribution designed manually after flatting by parameter temperature.

**Training algorithm.** The first step is to select appropriate parameters to design this probability distribution to obtain a virtual teacher network with completely correct outputs. The second step is to use the virtual teacher network to guide the training of the student network through knowledge distillation. The virtual teacher networks have a 100% accuracy and do not have interference from incorrect classification, so they can better guide the training of student networks.

## 4 EXPERIMENTAL RESULTS

We conducted experiments on the static dataset CIFAR100 and Tiny-imagenet and on the neuromorphic dataset DVS-Gesture to verify the effectiveness of this framework. Firstly, we analyze the impact of different pruning ratios of the teacher network on the performance of the student network. Then we verify the performance of knowledge distillation with a virtual teacher network. In addition, in order to validate the advantages of this re-KD method, we compare the experimental results of this framework with other SNN methods.

### 4.1 Experimental settings

We conduct the experiments on the server which is equipped with 16 cores Intel(R) Xeon(R) Gold 5218 CPU with 2.30GHz and 8 NVidia GeForce RTX 2080 Ti GPUs. The training of SNN is based on the spikingjelly framework, which is an SNN framework developed based on pytorch. Our experiments mainly focus on residual structures such as Resnet and WideResnet. In this experiment, the teacher network adopts the same structure as the student network in order to better verify the advantages of structure learning.

**Dataset CIFAR100 and Tiny-imagenet.** CIFAR100 is a commonly used static dataset, it has three channels RGB and the image size is 32*32. The CIFAR100 dataset has 100 classes, and each class has 500 training sets and 100 test sets. The image size of Tiny-imagenet is 64*64 and it has 200 classes, each class has 500 training sets and 50 test sets. Each image in both datasets has fine-labels and coarse-labels two labels. The dataset is relatively complex and can verify the performance of our proposed framework in a more realistic way. During training, we need to first encode the static images into spiking sequences and then input them into the SNN, here the first spiking neuron layer in the network is regarded as the encoding layer.

**Dataset DVS-Gesture.** DVS-Gesture is a neuromorphic dataset that includes 11 gestures for recognition. The dataset is stored in the form of events and needs to be integrated into frame data before use. During training, the spiking sequences data obtained can be directly input into SNN. The dataset has 2 channels and the image size is 128×128.

### 4.2 Evaluation on sparse-KD method

We use the SNN student network itself and the SNN student network with different pruning ratios as the teacher network and then analyze the experimental results. In this experiment, we used the spiking form of Resnet18, WRN16-4 and 5Conc-1FC ([6]) network structures to conduct the experiments.

For the CIFAR100 dataset, as shown in Table 1, using the SNN student network itself as its own teacher network for knowledge distillation improves the accuracy of the student network by 1.33% (Resnet18) and 2.46% (WRN16-4). The experiments also show that using models with different pruning ratios of the student network itself as the teacher network can also improve the accuracy of the student network. However, using a pruned model as a teacher network for knowledge distillation is more effective than using a non-pruned model as a teacher network. As can be seen in Table 1, for Resnet18, when using a model pruned at 0.1 as the teacher network, the accuracy of the student network can be improved by 1.49%, which is greater than 1.33%. For WRN16-4, when using a model pruned at 0.1 as the teacher network, the accuracy of the student network can be improved by 2.60%, which is greater

**Table 1: Test accuracies of sparse-KD on CIFAR100, DVS-Gesture and Tiny-imagenet.**

| Dataset | SNN Model | Teacher prune ratio | Teacher Acc. (%) | Student SNN ACC. (%) | Student KD ACC. (%) | Improvement(%) |
|---|---|---|---|---|---|---|
| CIFAR100 | Resnet18 | 0.1 | 70.69 | 71.30 | 72.79 | 1.49 |
| | | 0.3 | 71.10 | 71.30 | 72.83 | 1.53 |
| | | 0.5 | 70.18 | 71.30 | 72.61 | 1.31 |
| | | 0.7 | 65.73 | 71.30 | 73.01 | 1.71 |
| | | 0 | 71.30 | 71.30 | 72.63 | 1.33 |
| | WRN16-4 | 0.1 | 68.73 | 69.30 | 71.90 | 2.60 |
| | | 0.3 | 68.23 | 69.30 | 72.15 | 2.85 |
| | | 0.5 | 66.32 | 69.30 | 71.69 | 2.39 |
| | | 0.7 | 50.94 | 69.30 | 71.79 | 2.49 |
| | | 0 | 69.30 | 69.30 | 71.76 | 2.46 |
| Tiny-imagenet | Resnet18 | 0.1 | 54.96 | 54.96 | 56.92 | 1.96 |
| | | 0.3 | 54.96 | 54.96 | 56.75 | 1.79 |
| | | 0.5 | 55.02 | 54.96 | 56.57 | 1.61 |
| | | 0.7 | 54.79 | 54.96 | 56.78 | 1.82 |
| | | 0.9 | 35.81 | 54.96 | 56.61 | 1.65 |
| | | 0 | 54.96 | 54.96 | 56.67 | 1.71 |
| DVS-Gesture | 5Conv 1FC | 0.1 | 94.44 | 94.79 | 96.18 | 1.39 |
| | | 0.3 | 93.06 | 94.79 | 95.83 | 1.04 |
| | | 0.5 | 90.97 | 94.79 | 95.83 | 1.04 |
| | | 0.7 | 76.04 | 94.79 | 95.83 | 1.04 |
| | | 0 | 94.79 | 94.79 | 95.83 | 1.04 |

**Table 2: Test accuracies of default-KD on CIFAR100, DVS-Gesture and Tiny-imagenet.**

| Dataset | SNN Model | Student SNN ACC. (%) | Student KD ACC. (%) | Improvement(%) |
|---|---|---|---|---|
| CIFAR100 | Resnet18 | 71.30 | 72.10 | 0.80 |
| | WRN16-4 | 69.30 | 71.52 | 2.22 |
| | VGG16 | 65.49 | 65.95 | 0.46 |
| Tiny-imagenet | Resnet18 | 54.96 | 59.31 | 4.35 |
| DVS-Gesture | 5Conv 1FC | 94.79 | 95.13 | 0.34 |

than 2.56%. As the pruning ratio increases, the improvement of the accuracy of the student network generally decreases.

For the Tiny-imagenet dataset, as shown in Table 1, using the SNN Resnet18 student network itself as its own teacher network for knowledge distillation improves the accuracy of the student network by 1.71%. The teacher network pruned at 0.1 imporves the accuracy of the student network better by 1.96%. When the teacher model is pruned at 0.1 and the accuracy is only 35.81%, it still can conduct knowledge distillation training on student networks. To better compare the effectiveness of knowledge distillation using different pruning ratios and unpruned teacher networks, we visualize a line graph of test accuracy. As shown in Figure 2, the blue line represents the accuracy of the student network before any pruning, while the orange line represents the accuracy of the student network using the unpruned model as the teacher network. The other lines above the orange line represent the accuracy of the student network using different ratios of pruning as the teacher network.

For the DVS-Gesture dataset, as shown in Table 1, using the SNN student network itself as its own teacher network for knowledge distillation improves the accuracy of the student network by 1.04%. When pruning the teacher network at 0.1, the improvement effect on the student network is more obvious, and it improves by 1.39%.

Using a pruned student network or the student network itself for knowledge distillation is different from the general perception of knowledge distillation. The accuracy of the pruned student network may be slightly lower than that of the student network, but experimental results show that it can still improve the accuracy of the student network. This shows that the pruned model, by reducing some of the interference from redundant connections, can better transfer effective hidden knowledge to guide the training of the student network.

## 4.3 Evaluation on teacher default-KD method

We use a virtual teacher network to perform knowledge distillation. The experiment is based on the spiking form of Resnet18, WRN16-4, and VGG16 network structures. The virtual teacher network is a

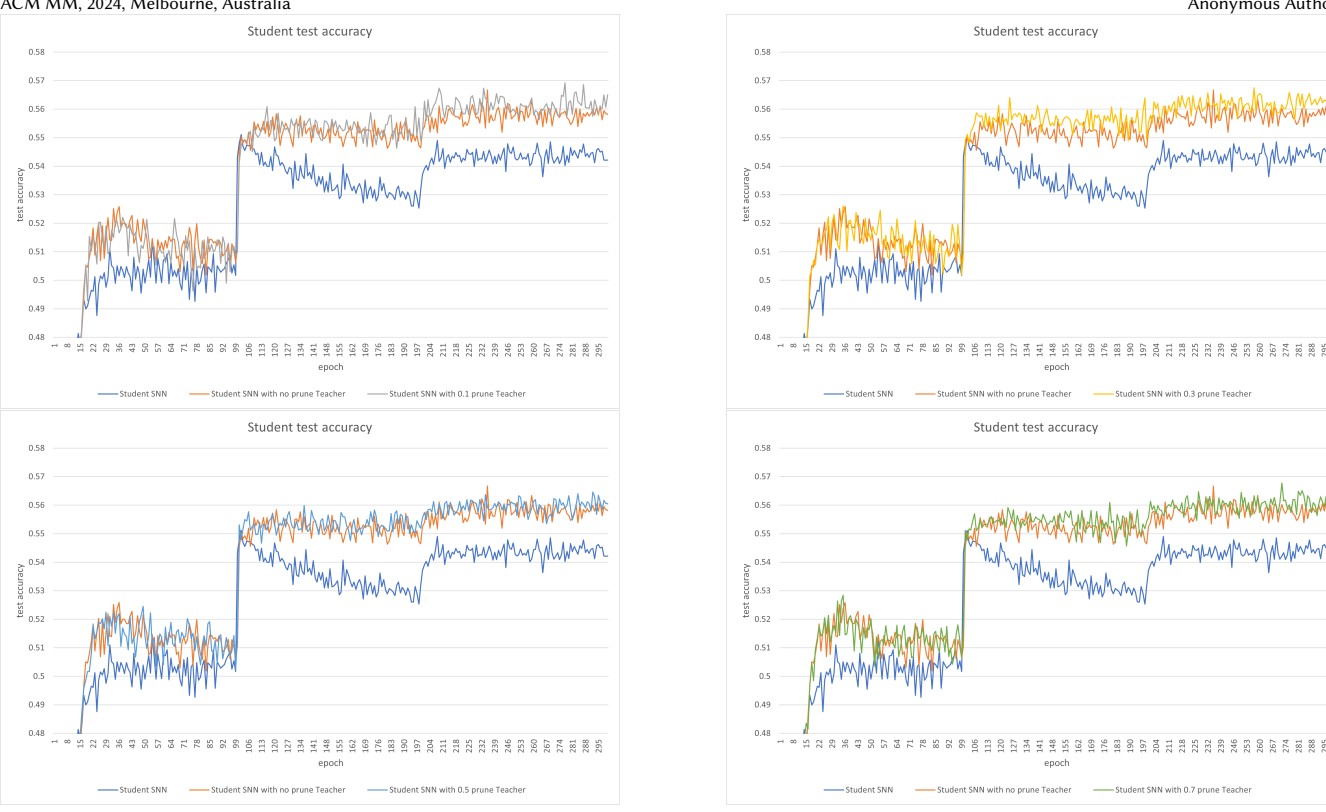

Figure 2: Comparison of test accuracy of student SNNs under different pruning ratios of teacher networks.

Table 3: Summary comparison of classification accuracies with other spiking based models

| Dataset | Method | SNN Architecture | SNN Acc.(%) | timestep |
|---|---|---|---|---|
| CIFAR100 | RMP-snns [9] | VGG16 | 70.93 | 2048 |
| | | Resnet20 | 67.82 | |
| | Hybrid [20] | VGG11 | 67.87 | 125 |
| | TSC [8] | VGG16 | 70.97 | 1024 |
| | | Resnet20 | 68.18 | |
| | Opt. [4] | VGG16 | 70.55 | 400 |
| | | Resnet20 | 69.82 | |
| | **Proposed sparse-KD** | ResNet18 | 73.01 | 4 |
| | **Proposed default-KD** | ResNet18 | 72.10 | 4 |
| Tiny-imagenet | LTL [28] | Resnet20 | 56.28 | 16 |
| | BNTT [14] | VGG11 | 57.8 | 30 |
| | **Proposed sparse-KD** | Resnet18 | 56.92 | 4 |
| | **Proposed default-KD** | Resnet18 | 59.31 | 4 |
| DVS-Gesture | PLIF [6] | c128k3s1-BN-PLIF-MPk2s2*5-DPFC512-PLIF-DP-FC110-PLIF-APk10s10 | 97.57 | 20 |
| | STBP-TdBN [32] | Resnet17 | 96.87 | 40 |
| | SLAYER [22] | 8 layers | 93.64 | 25 |
| | Com. [10] | Input-MP4-64C3-128C3-AP2-128C3-AP2-256FC-11 | 93.40 | 25 |
| | **Proposed sparse-KD** | 5Conv 1FC | 96.18 | 16 |
| | **Proposed default-KD** | 5Conv 1FC | 95.13 | 16 |

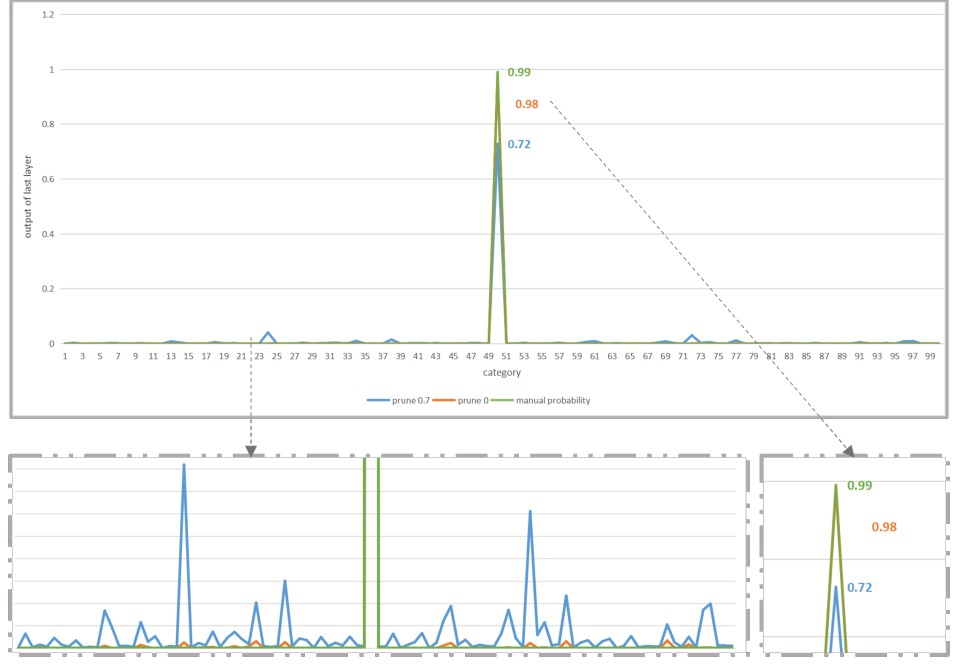

Figure 3: The visualization of the output distributions of the teacher network's last layer in different methods.

probability distribution designed by humans, with an accuracy of 100%.

The effectiveness of using virtual teacher networks for different models varies, as shown in Table 2. On the CIFAR100 dataset, using knowledge distillation with a virtual teacher network on a WRN16-4 model improves the accuracy of the student network by 2.22%. For the Resnset18 model, the student network's accuracy is improved by 0.80%. On the Tiny-imagenet dataset, the student network's accuracy is improved by 4.35% which is greater than sparse-KD method. On the DVS-Gesture dataset, the student network's accuracy is also slightly improved over 0.34%.

Using a virtual teacher network to perform knowledge distillation on a student network is an effective way to improve the accuracy of the student network. The virtual teacher network does not have any output errors and can better guide the training of the student network. Since the training of a greater teacher SNN is difficult, this method is also suitable for situations where it is difficult to find a model with better performance as a teacher network for knowledge distillation.

### 4.4 Visualization Analysis

We visualize the output distributions of the teacher network's last layer in different methods. For sparse-KD, as shown in Figure 3, we can observe that the pruned teacher network's output (blue line) is relatively smoother compared to the unpruned teacher network (orange line), and it contains inter-class information (e.g., class 24 "clouds" and class 50 "mountains"). As for default-KD, we can see that the probability distribution designed by the virtual teacher network (green line) is mostly consistent with that of the original teacher network (orange line).

### 4.5 Performance Comparison with Other Methods

As shown in Table 3, in order to better analyze the effectiveness of the method, we compare it with some existing methods. For the CIFAR100 dataset, the sparse-KD and default-KD methods using a pruned model with a pruning rate of 0.3 as the teacher network on the Resnet model have an accuracy of 72.83% and 72.10%, respectively, which is better than the Rmp-snns [9], Hybrid [20], Opt. [4] and TSC [8] methods. For the Tiny-imagenet, the accuracy of our method is better than LTL [28] with less time steps. For the DVS-Gesture dataset, the sparse-KD method using a pruned model with a pruning rate of 0.1 as the teacher network on the 5Conv-1FC structure has an accuracy of 96.18%, and the default-KD method has an accuracy of 95.13%, both of which are better than the SLAYER [22] and Com. [10] methods. Compared to the PLIF [6] and STBP-TdBN [32] methods, the accuracy is slightly lower, but the time step used is 16, which is shorter than them.

## 5 CONCLUSION

Inspired by the biological plausibility of neural systems in human brain, this paper proposed efficient structure learning methods with reverse knowledge distillation for SNNs. This kind of method could help to build a deep but sparse structure under the guidance of the pruning method which could not only discard the redundancy of the complex spiking neural dynamics but also save power consumption and memory storage in SNNs.

Considering the abnormal KD cases such as the proposed sparse-KD and teacher-default KD methods, experimental results showed that we can expand our work to broader conditions in SNNs especially when the teacher model is weak. It also showed that the

proposed models not only get good performance on a relatively large dataset (CIFAR100 and Tiny-imagenet) but are also suitable for the dynamic scene (DVS-Gesture). Under strict timesteps, the proposed method can help SNNs get good performance compared to other spiking based models. That is to say, the proposed methods could give full play to the advantages of low power consumption of SNNs.

In our future work, we will expand the structure learning methods to utilize the advantages of spiking signals, not limited by the proposed two teacher-weak conditions, we will consider more situations when the teacher model is ill or disabled and evaluate them on more types of datasets.

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
