# OpenReview forum: "Reversing Structural Pattern Learning with Biologically Inspired Knowledge Distillation for Spiking Neural Networks"
_acmmm.org/ACMMM/2024/Conference — MM2024 Poster_

### Official Review · Reviewer_3ePt · 2024-05-14

**Rating:** 5
**Confidence:** 4

**Summary:**

This paper proposes an evolutionary-based structure construction method for constructing more reasonable and high performance SNNs. The reverse-KD framework explores a method of knowledge distillation that utilizes a network with a sparse structure or a virtual network as the teacher network, including two approaches: sparse-KD and teacher default-KD. This method offers a new approach to knowledge distillation in the SNN field by training student networks to enhance their performance under resource-constrained conditions.

**Strengths:**

The paper introduces a novel method for training SNNs through reverse knowledge distillation. It utilizes a weak teacher network to train a student network, effectively enhancing the student model's performance. This approach is particularly beneficial in scenarios where it's challenging to find a higher-performing teacher network under resource-constrained conditions. It leverages the low-power and biologically interpretable advantages of SNNs and presents a new way of absorbing hidden knowledge in teacher networks through learning.

**Limitations:**

The paper has some shortcomings as follows:
1. In the paper, the motivation behind proposing this reverse knowledge distillation method is not sufficiently clear. It needs to elaborate on the benefits of introducing this distillation approach for training SNNs.
2. The paper fails to provide a reasonable explanation for the biological basis mentioned in the abstract. Are there any related references that could elucidate this aspect?
3. There are some spelling and grammar errors in the paper. Some of the illustrations are also rather blurry.

**Suitability:**

3

---

### Official Review · Reviewer_fNfK · 2024-05-16

**Rating:** 5
**Confidence:** 2

**Summary:**

This paper presents a novel method for structuring Spiking Neural Networks (SNNs) using biologically inspired knowledge distillation. The method proposes an evolutionary-based structure construction to reduce redundancy and dynamically optimize SNNs. It integrates knowledge distillation with connection pruning, allowing SNNs to learn from a teacher model and search for a deep yet sparse network topology. Experimental results show the method enhances performance while reducing connection redundancy.

**Strengths:**

This paper has a new insight on the training process that jointly considering structural learning and distillation learning.

This paper introduces a structure construction method that dynamically optimizes synaptic connections in SNNs, making them more biologically plausible and efficient.

Experimental results on datasets like CIFAR100, Tiny-imagenet, and DVS-Gesture demonstrate that the proposed method enhances performance while reducing connection redundancy.

**Limitations:**

In Section 3.2, why the teacher model needs to be sparse is not clearly explained. The authors claimed that removing these redundant connections makes the network structure more robust. While the authors are not mentioning what pruning methods they are using.

In Section 3.3, the authors introduce a way to construct a teacher model that is 100% correct. I am confused about how to achieve this goal using the technique in Section 3.3 and how to integrate this technique into the distillation framework.

Texts in Figures 2 and 3 are not readable.

One extra question: if using a sparse teacher model for distillation, will the student SNN be sparse?

**Suitability:**

2

---

### Official Review · Reviewer_b5wT · 2024-05-23

**Rating:** 5
**Confidence:** 3

**Summary:**

This paper proposes sparse-KD and teacher default-KD to enhance the performance of spiking neural networks, and conducts extensive experiments on static and neuromorphic datasets to validate the effectiveness of the proposed methods.

**Strengths:**

Inspired by biological neural systems, sparse-KD attempts to construct a deep but sparse SNN by using the pruned model as the teacher to guide the training of the unpruned model. The teacher default-KD allows for direct distillation without additional teacher models by manually defining soft labels. The authors conducted extensive experiments on three datasets to demonstrate the advantages of the proposed methods in improving the performance of SNNs.

**Limitations:**

1. In sparse-KD, the authors prune the trained SNN as the teacher and take the SNN model with the same structure as the student for distillation (line 352). However, it is not clear from the paper whether the student model is randomly initialized or has the same weights as the trained but unpruned model, and it is hoped that the authors will clarify this.
2. Table 1 shows that using a model with a pruning rate of 0 as the teacher can also improve the performance of the SNN model. The teacher and student at this point have exactly the same architecture and parameters, and their output logit is the same. In this case, a distillation loss of 0 would not cause the error gradient to propagate, so why would it improve performance?
3. In the experimental section, the authors should compare the proposed method with recent SOTA methods.

**Suitability:**

2

---

### Official Review · Reviewer_Rpsh · 2024-05-24

**Rating:** 2
**Confidence:** 3

**Summary:**

This paper designs a novel method for improving the efficiency and performance of SNNs. Here are the key points from the paper:

 **Proposed Solution**: The paper proposes an evolutionary-based structure construction method that combines knowledge distillation and connection pruning. This approach aims to optimize synaptic connections in SNNs, aligning them more closely with the efficient neural topologies observed in the human brain.

**Benefits**: By integrating these methods, the structure of SNNs can be refined to learn from teacher models and maintain efficient sparse network topology. This reduces computational redundancy and enhances performance.

**Experimental Results**: Tests on CIFAR100, Tiny-Imagenet, and DVS-Gesture datasets demonstrate that the proposed learning method achieves significant performance improvements while reducing connection redundancy.

**Strengths:**

1. Dynamic Structure Learning: The approach allows SNNs to dynamically adjust their structure during the learning process.
2. Reverse Knowledge Distillation: Unlike traditional knowledge distillation, this method trains the student SNNs by learning from the teacher ANNs in a reverse manner, focusing on structural efficiency.

**Limitations:**

The main limitation is there's no experiments on large dataset like ImageNet-1K. While there are already lots of SNNs working on ImageNet with direct training method, this new KD method could make a comparison.

**Suitability:**

2

---

### Official Review · Reviewer_G23S · 2024-05-25

**Rating:** 4
**Confidence:** 3

**Summary:**

This paper proposes an efficient structure learning methods for Spiking Neural Networks (SNNs) with reverse knowledge distillation. By integrating knowledge distillation and connection pruning techniques, the synaptic connections within SNNs can be dynamically optimized to achieve an optimal state. This allows the SNN structure to not only learn from a teacher model but also to explore deep yet sparse network topologies. Experiments conducted on datasets such as CIFAR100, Tiny-ImageNet, and DVS-Gesture demonstrate that the proposed structure learning method achieves competitive performance while reducing connection redundancy. This innovative dynamic approach to structure learning in SNNs helps bridge the gap between deep learning and bio-inspired neural dynamics.

**Strengths:**

1. The authors introduce advanced techniques such as structural learning and knowledge distillation into Spiking Neural Networks (SNNs), making this a novel, interesting, and worthwhile topic to explore.
2. The motivation, relevant background, and methodological details are presented by the authors with great clarity and thoroughness.

**Limitations:**

1. Although the authors conducted experiments on image datasets (i.e., CIFAR100 and Tiny-ImageNet) and an event dataset (DVS-Gesture), these datasets are relatively small in scale. Specifically, for the DVS-Gesture dataset, most existing methods already achieve over 0.95 accuracy, making it challenging to differentiate between methods. It is suggested that the authors experiment with larger-scale event datasets, such as the ASL-DVS dataset.
2. Further ablation studies are needed, for instance, on the hyperparameter 𝑎 in the loss function, as different values might impact the final performance.
3. The authors should report the inference time or algorithmic complexity of different methods.
4. One of the main advantages of Spiking Neural Networks is their low power consumption when deployed on neuromorphic chips. Currently, most algorithms include power consumption analyses. The authors should theoretically analyze the feasibility of deploying their operations on neuromorphic chips and provide theoretical or experimental statistics on power consumption.
5. The authors are encouraged to further improve the writing of the paper, including the text, figures, and tables. Additionally, the official limit allows for up to 8 pages of main content, so the authors have room to include more details in the paper.

**Suitability:**

2

---

### Meta-Review · Area_Chair_BN7i · 2024-06-26

**Recommendation:** Accept (Poster)
**Confidence:** 5

**Metareview:**

From the final rating, four reviewers gave a Weak Accept, while one gave a borderline reject. Overall, there is broad agreement that this paper presents a novel method for improving the efficiency and performance of SNNs. It introduces a structure construction method that dynamically optimizes synaptic connections in SNNs, making them more biologically plausible and efficient. Experimental results on datasets like CIFAR100, Tiny-ImageNet, and DVS-Gesture demonstrate that the proposed method enhances performance while reducing connection redundancy. Therefore, I recommend the acceptance.